# Increased Risk of Acquisition of New Delhi Metallo-Beta-Lactamase-Producing Carbapenem-Resistant Enterobacterales (NDM-CRE) among a Cohort of COVID-19 Patients in a Teaching Hospital in Tuscany, Italy

**DOI:** 10.3390/pathogens9080635

**Published:** 2020-08-05

**Authors:** Andrea Davide Porretta, Angelo Baggiani, Guglielmo Arzilli, Virginia Casigliani, Tommaso Mariotti, Francesco Mariottini, Giuditta Scardina, Daniele Sironi, Michele Totaro, Simona Barnini, Gaetano Pierpaolo Privitera

**Affiliations:** 1Department of Translational Research and New Technologies in Medicine and Surgery, University of Pisa, 56123 Pisa, Italy; angelo.baggiani@unipi.it (A.B.); g.arzilli3@studenti.unipi.it (G.A.); v.casigliani@studenti.unipi.it (V.C.); 29397007@studenti.unipi.it (T.M.); francesco.mariottini@med.unipi.it (F.M.); g.scardina@studenti.unipi.it (G.S.); daniele.sironi@outlook.it (D.S.); m.totaro2@studenti.unipi.it (M.T.); gaetano.privitera@unipi.it (G.P.P.); 2University Hospital of Pisa, 56123 Pisa, Italy; 3Bacteriology Unit, University Hospital of Pisa, 56123 Pisa, Italy; s.barnini@ao-pisa.toscana.it

**Keywords:** COVID-19, health care related infections, surveillance, New Delhi metallo-beta-lactamase

## Abstract

We describe the epidemiology of New Delhi Metallo-Beta-Lactamase-Producing Carbapenem-Resistant Enterobacterales (NDM-CRE) colonization/infection in a cohort of COVID-19 patients in an Italian teaching hospital. These patients had an increased risk of NDM-CRE acquisition versus the usual patients (75.9 vs. 25.3 cases/10,000 patient days). The co-infection significantly increased the duration of hospital stay (32.9 vs. 15.8 days).

## Short Communication

Secondary bacterial infections in patients with COVID-19 have already been described and have an impact on their clinical management, and outcomes [1,2]. Healthcare-associated infections (HCAI) are therefore a cause of concern in this group of patients.

In the months of March and April 2020, hospitals in Tuscany experienced a sudden influx of COVID-19 patients, a large proportion requiring intensive care, in a short period of time. In the Pisa Teaching Hospital, the organizational structure, including the establishment of COVID designated wards, and patient management protocols were modified to address the clinical needs of the patients, as well as the concern for the protection of healthcare workers from infection by SARS-CoV-2 [3]. However, Tuscany hospital services were already facing a challenge represented by the diffusion of multidrug-resistant organisms (MDRO), especially New Delhi metallo-beta-lactamase-producing carbapenem-resistant Enterobacterales (NDM-CRE) [4,5]. This issue was addressed by linking the already existing HCAI surveillance system with the epidemic surveillance protocols for COVID-19. We describe here the diffusion of NDM-CRE infection/colonization among COVID-19 cases in a large teaching hospital in Tuscany, Italy, at the peak of the pandemic in the region.

A database was established containing a unique identifier and the following information for each patient: sex, age, date of admission, specialty of hospitalization, if transferred from another hospital, reason for admission, date of first positive sample for SARS-CoV-2, date of first positive for MDRO, blood culture positive for MDRO if received, and date and type of discharge (death, discharge at home, or transfer to another hospital). All patients admitted before 31 May 2020, regardless of reason for hospitalization, with at least a positive sample for SARS-CoV-2 were included. A descriptive analysis of data was performed.

In the period from 4 March to 31 May 2020, 331 patients were admitted to our hospital, either for clinical COVID-19 or for other reasons, with a positive sample for SARS-CoV-2. Forty-three of them had been transferred with a diagnosis of COVID-19 from other hospitals. The mean age was 67.4 years (median 70; IQR, 56.5–80; range, 18–99 years), and 216 of 331 (65.25%) were male. The wards of admission were the COVID specific intensive care unit (ICU) for 61 patients (of which 31 transferred from other hospitals), COVID specific non-ICU for 245 patients (of which 12 transferred from other hospitals), non-COVID non-ICU wards for 23, and non-COVID ICU for two.

Of the 331 patients, 228 (68.8%) were discharged with a clinical recovery, 22 (6.6%) were transferred to another hospital, and 81 died (24.5%).

Of the 81 deaths, 12 occurred in patients admitted for life-threatening conditions such as ictus and who tested positive for SARS-CoV-2 during hospitalization.

In the wake of the public health alert in the month of February, COVID-19 designated wards of internal medicine, pneumology, infectious diseases, and ICU were established. A number of patients were transferred among wards during their stay, and for the purposes of this study, ICU patients are defined as patients with at least a day of presence in an ICU: 81 patients (24.5%) stayed in the ICU during hospitalization. For the remaining patients, we refer to the specialty of admission. A total of 282 patients were hospitalized for COVID-19 related symptoms, while 49 were admitted with other diagnoses, with a positive result for SARS-CoV-2.

Excluding patients who died during hospitalization, for the remaining 250 patients, the mean duration of stay was 18.7 days and the median 15 days. We observed that ICU patients (n = 51, excluding dead patients) had a mean hospitalization of 31 days and a median of 24 days, while in non-ICU patients (n = 199 discharged or transferred to another hospital), the mean was 15.6 days and the median was 13 days. ICU patients who died (n = 30) had a mean duration of hospitalization of 17.6 days and a median of 11.5 days, while in the 51 non-ICU patients who died in hospital, the mean duration of stay was 10 days and the median was 5 days.

HCAI hospital protocols require screening by rectal swab of all patients for the presence of MDROs; in the population of COVID-19 patients, at least one sample for an MDRO was obtained for 47 patients (14.2%; 34 males; mean age 69.1 years; median 72; IQR, 62.5–77; range, 25–96 years). Five patients were already colonized at admission (three NDM-CRE organisms and two Carbapenemase producing *Klebsiella* spp. (KPC)). The number of patients with colonization/infection by two MDROs was four (8.5%), and a patient was hospitalized who was already colonized by KPC and acquired during the hospital stay both NDM-CRE and *Acinetobacter baumannii*. During hospital stay, six patients were colonized by *Acinetobacter baumannii*, two by KPC, and 40 by NDM-CRE. In Figure 1, the number of COVID patients present in the hospital for each day, and the date of detection of NDM-CRE cases, are given.

Considering only the 43 patients with at least one sample positive for NDM-CRE, their mean age was 69.8 years, median 72 years (IQR, 65–77.5; range, 25–96 years), and 31 were males (72.1%). Of all the NDM-CRE patients, three were positive at admission, while 40 acquired NDM-CRE colonization/infection during their hospital stay, with a median time of acquisition of 17 days (mean 15 days). Ten NDM-CRE patients (9 males, mean age 72.5; median 74; IQR, 70.25–76.5; range, 60–83 years) developed a blood infection during their stay in hospital. Three infected patients died during hospitalization (3/9 = 33.3%). More than one third of patients with at least a day of permanence in an ICU were NDM-CRE cases (n = 31; 38.3%).

The rate of NDM-CRE cases per 10,000 hospital days was 75.9, while in the same wards, for the previous year, the rate was 25.3. Even taking into account the variations in case mix between the two time intervals (e.g., the main COVID-19 ICU is normally a post-surgical one), this difference is a cause for concern. Considering mean duration of hospitalization, COVID + NDM-CRE patients had a longer duration of stay compared to NDM-CRE patients in the previous year (40.2 vs. 21.9 days) and this can be explained by the features of COVID clinical course.

Student’s *T* test was used to compare different groups of patients. The interval between admission and the first positive sample for NDM-CRE was not significantly different between the 29 ICU patients (median time of acquisition 17 days) and the 11 non-ICU patients (median time of acquisition 14 days).

Lethality rates between NDM-CRE and non-NDM-CRE patients (25.6% vs. 24.7%) were not significantly different.

The distribution of duration of hospitalization by specialty and NDM-CRE status is given in Table 1. Excluding the 81 patients who died, NDM-CRE patients had a significantly longer mean hospital stay than non-NDM-CRE patients, respectively 40.2 days and 15.8 days (*p* = 0.0001), and the same observation can be made if we consider ICU patients alone (NDM-CRE average stay 41.3 vs. non-NDM-CRE 22.9; *p* = 0.0012) and non-ICU patients (NDM-CRE average stay 32.3 vs. non-NDM-CRE 14.8; *p* = 0.0001).

The sudden influx of COVID-19 patients in our hospital put an additional strain on the challenge posed by the presence and diffusion of MDROs, especially NDM-CRE, in the hospital environment.

Coinfection with SARS-CoV-2 and MDROs has a significant impact on the clinical course and on the outcome of hospitalized patients, as shown by the significantly increased length of stay for NDM-CRE patients.

Colonization by contact transmitted pathogens in the hospital environment has been often linked to procedures and environment [6]. Further analysis of the clinical data is needed to determine whether the high rate of diffusion of NDM-CRE among COVID-19 patients is to be attributed to their features or to their clinical and organizational management, in respect to wards, patients, and healthcare workers’ safety, but it already appears that there is the need for a strategy and guidelines for the management of the simultaneous risk for the two infections.

## Figures and Tables

**Figure 1 pathogens-09-00635-f001:**
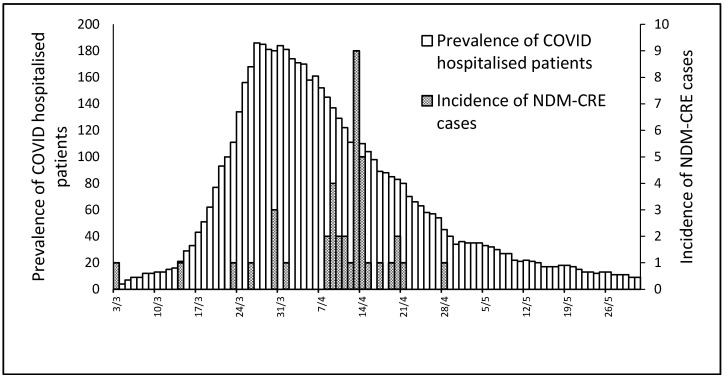
Prevalence of hospitalized COVID-19 patients and incidence of NDM-CRE cases.

**Table 1 pathogens-09-00635-t001:** Duration of hospitalization by specialty and NDM-CRE status.

	NDM Patients (Mean; Median Days)	Non-NDM Patients (Mean; Median Days)	*T*-Test (*p*)
ICU	23 patients (41.3; 38)	28 patients (22.9; 22)	0.0012
NON-ICU	9 patients (32.3; 32)	190 patients (14.8; 13)	0.0001
Total	32 patients (40.2; 34.5)	218 patients (15.8; 14)	0.0001

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
