# Peer review of "Increased Risk of Acquisition of New Delhi Metallo-Beta-Lactamase-Producing Carbapenem-Resistant Enterobacterales (NDM-CRE) among a Cohort of COVID-19 Patients in a Teaching Hospital in Tuscany, Italy"

_pathogens, 2020, doi:10.3390/pathogens9080635_

Round 1

Reviewer 1 Report

In the current study, the authors described the increased risk of acquisition of NDM-producing isolates among a cohort of COVID-19 patients in Italy. The study is well presented.

Comments:

Line 37: correct to ‘culture’.

Line 73: correct to ‘KPC’.

Author Response

Thank you for your attention,

we made the corrections in the text.

We took note of your (and of the other reviewers aswell) request about the results part.
We specified that the comparison among duration of hospitalisation was tested for significativity by using the T student test.

Andrea

Reviewer 2 Report

Overall, the paper is well written.  There are a few wording errors which can be improved to increase the clarity of the manuscript.  Suggested wording changes are found in notes added to the attached file.

Very little information is provided as to the statistics that were used to compare different groups of patients.  A paragraph briefly explaining the statistics would be helpful for the readers to understand what has been done.  

Author Response

We are grateful for your suggestions in improving the quality of the text, suggestions that have been promptly incorporated in the resubmitted version.

On the topic of the results, we specified that the T-Student test was used to determine significativity of the observed differences between groups of patients.

Andrea

Reviewer 3 Report

Porretta et al. describe the acquisition of NDM-CRE by COVID-19 patients in an Italian teaching hospital. The short communication is topical and I believe the results of the study are interesting, but I have several comments/suggestions that may improve the presentation of the manuscript.

In the abstract, the authors state that a cohort of COVID-19 patients had an increased risk of NDM-CRE acquisition versus non-COVID-19 patients (75.9 vs. 25.3 cases/10.0000 patient days). However, the rate of acquisition for non-COVID-19 patients was derived from an analysis of patients in the same wards on the previous year. Do the authors have a sense of whether COVID-19 status was an independent predictor of NDM-CRE acquisition, or alternatively, if the spread of NDM-CRE was more closely related to infection control measures during the pandemic? The authors state that further analysis is needed to determine the cause of the NDM-CRE spread, but it may be beneficial to put their preliminary assessment into the manuscript (especially considering the title of the manuscript seems to imply that COVID-19 may increase the risk of CRE acquisition). Also, do the authors have the rates of NDM-CRE acquisition for non-COVID-19 patients during March and April 2020?

The use of statistics in the manuscript is a little inconsistent. On lines 94 – 95, the authors state that the lethality rates between NDM-CRE and non-NDM-CRE patients were not significantly different and do not provide a p value. Then in lines 96 – 100, the authors disclose several p values but do not mention the statistical test that was used. I recommend adding the statistical method that was utilized in the first parenthetical comment that lists a p value.

It is a little difficult at times to keep track of all the patient information that is discussed. If the Journal permits the use of one figure and one table for a short communication, it will be helpful to have a concise table that lists basic information such as: the number of COVID-19 patients, the number of COVID-19 patients that tested positive for NDM-CRE, descriptive statistics for the NDM-CRE patients with COVID-19, etc.

The manuscript is relatively well-written, but there are numerous grammatical mistakes and awkward phrases that detract from the flow of the short communication. I recommend that a native English speaker read the manuscript and offer recommendations for optimizing the writing. For example, some suggested edits to the paragraph contained in lines 68 – 76 include:

  • “… in the population of COVID-19 patients, 47 of the patients tested positive for MDROs.”
  • “There were 4 patients that were colonised/infected by two MDROs…”
  • “During their hospital stay, 6 patients became colonized with Acinetobacter baummannii…”

Author Response

Thank you for your in-depth analysis and observation on our paper.

Your questions are indeed quite challenging and we hope to have included in the revised draft some clearer description and conclusion regarding our observations.

1.- The "control" group, represented by the previous year's patients in the same wards suffers from a clear difference in the case mix when compared with the COVID-19 patients of our cohort. Rates of acquisition for non-COVID patients in the period under observation aren't available at the moment because at present data on duration of stay of non covid patients for the first months of 2020 are unavailable.

2.- We included in the result section a reference to the T-Student as the statistical test we used for the results in lines 94-95.

3.- We included a table with the information suggested

Finally, we incorporated the edits you suggested and did our best to improve the quality of the text.

We think the main point of our paper, i.e. that  the management of COVID-19 patients requires a special attention also to the management of the risk of acquisition of NDM-CREs remains valid, regardless of the reason for this increase.
Colonisation of hospitalised patients by contact transmitted pathogens occurs independently from the immune status of the patient themselves, and is usually linked to the procedures performed in the ward.

Thus we are justified in advancing the hypotesis that the observed high rate of transmission of NDM-CRE could be more closely associated to the change in procedures during the pandemic rather than to SARS-CoV-2 infection itself.

We are planning to widen the analysis to clinical data and to all the hospital population of Regione Toscana, but data will not be available until the months of October, while we wished to be able to raise attention to this issue earlier.

Andrea